# Cross-View Mutual Learning for Semi-Supervised Medical Image Segmentation

### Song Wu
School of Computer Science and Engineering, University of Electronic Science and Technology of China
Chengdu, Sichuan, China
songwu.work@outlook.com

### Xiaoyu Wei
School of Computer Science and Engineering, University of Electronic Science and Technology of China
Chengdu, Sichuan, China
weixiaoyu146@gmail.com

### Xinyue Chen
School of Computer Science and Engineering, University of Electronic Science and Technology of China
Chengdu, Sichuan, China
martinachen2580@gmail.com

### Yazhou Ren*
School of Computer Science and Engineering, University of Electronic Science and Technology of China
Chengdu, Sichuan, China
Shenzhen Institute For Advanced Study, University of Electronic Science and Technology of China
Shenzhen, Guangdong, China
yazhou.ren@uestc.edu.cn

### Jing He
Faculty of Medicine & Biomedical Sciences, The University of Queensland
Brisbane, QLD, Australia
Jing.he@uq.edu.au

### Xiaorong Pu
School of Computer Science and Engineering, University of Electronic Science and Technology of China
Chengdu, Sichuan, China
Shenzhen Institute For Advanced Study, University of Electronic Science and Technology of China
Shenzhen, Guangdong, China
puxiaor@uestc.edu.cn

## Abstract

Semi-supervised medical image segmentation has gained increasing attention due to its potential to alleviate the manual annotation burden. Mainstream methods typically involve two subnets, and conduct a consistency objective to ensure them producing consistent predictions for unlabeled data. However, they often ignore that the complementarity of model predictions is equally crucial. To realize the potential of the multi-subnet architecture, we propose a novel cross-view mutual learning method with a two-branch co-training framework. Specifically, we first introduce a novel conflict-based feature learning (CFL) that encourages the two subnets to learn distinct features from the same input. These distinct features are then decoded into complementary model predictions, allowing both subnets to understand the input from different views. More importantly, we propose a cross-view mutual learning (CML) to maximize the effectiveness of CFL. This approach requires only modifications to the model inputs and supervisory signals, and implements a heterogeneous consistency objective to fully explore the complementarity of model predictions. Consequently, the aggregated predictions can effectively capture both consistency and complementarity across two subnets. Experimental results on three public datasets demonstrate the superiority of CML over previous SoTA methods. Code is available at https://github.com/SongwuJob/CML.

*Corresponding author.

## CCS Concepts

• **Computing methodologies** → **Computer vision**; • **Theory of computation** → **Semi-supervised learning**.

## Keywords

Semi-supervised learning, Co-training, Mutual learning.

**ACM Reference Format:**
Song Wu, Xiaoyu Wei, Xinyue Chen, Yazhou Ren, Jing He, and Xiaorong Pu. 2024. Cross-View Mutual Learning for Semi-Supervised Medical Image Segmentation. In *Proceedings of the 32nd ACM International Conference on Multimedia (MM '24), October 28–November 1, 2024, Melbourne, VIC, Australia.* ACM, New York, NY, USA, 9 pages. https://doi.org/https://doi.org/10.1145/3664647.3680699

## 1 Introduction

Accurate identification of tissue structures or lesion regions is an essential task in medical image processing, and has achieved remarkable strides with the advance of deep learning [11, 12, 37, 40]. However, most current approaches generally rely on a large amount of meticulously annotated data, which is quite time-consuming, requiring expertise [8, 10]. To alleviate tedious manual labeling, semi-supervised medical image segmentation (SSMIS) methods [7, 17, 28, 32] have gained prominence, where they require only a limited amount of labeled data, and utilize abundant unlabeled data to achieve satisfactory effectiveness.

In SSMIS, the effective utilization of unlabeled data is an always-on topic [35]. Mainstream methods typically involve two similar networks, and conduct pseudo-label supervision [23] or consistency regularization [13] to make two subnets producing consistent predictions. Mean Teacher (MT) is a typical framework and propels a series of SSMIS works [19]. One of the most representative works is FixMatch [18], which employs the Teacher network to generate pseudo-labels on weakly-augmented unlabeled images, and

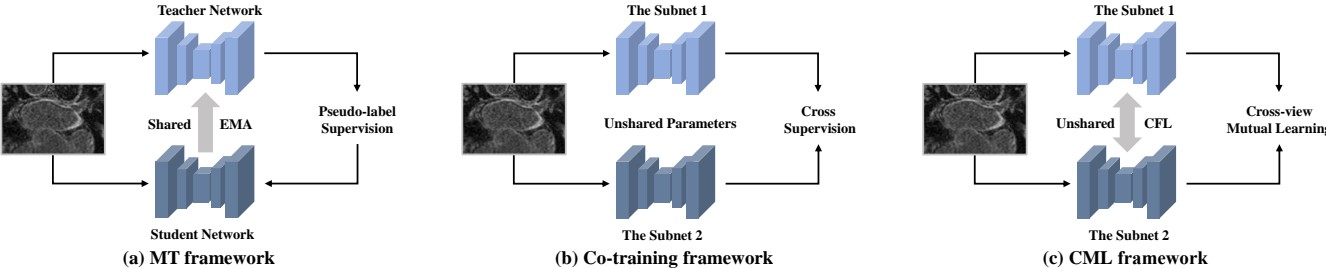

**Figure 1: A brief description of the process for Mean Teacher (MT), Co-training, and the proposed CML frameworks, respectively.**

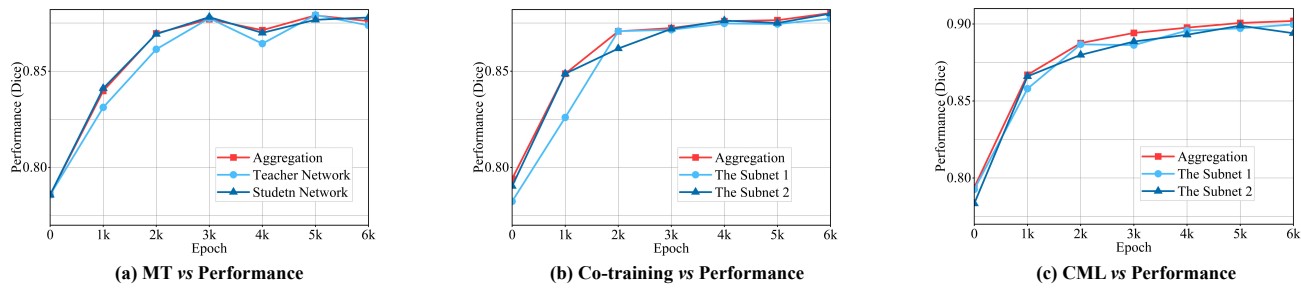

**Figure 2: The training process for MT, Co-training, and CML frameworks on LA dataset using 10% labeled data. (a) Teacher network, limited by EMA, exhibits performance variation consistent with Student network. (b) Co-training framework makes the two subnets presenting an adversarial rise, but the implementation of cross supervision gradually degrades them into the same view. (c) The proposed CML encourages the two subnets to reason the same input from different views, and the aggregated predictions benefited from complementary semantics achieve superior effectiveness.**

supervises the Student network learning the strongly-augmented version. Based on this, BCP [1] copy-pastes labeled and unlabeled data bidirectionally for alleviating empirical distribution mismatch. Despite the superior results achieved by these methods, the limitation imposed by Exponential Moving Average (EMA) hinders the Teacher network to reason individually about inputs, which wastes the potential of the multi-subnet architecture [25].

The co-training framework is a promising method to this end, which allows the two subnets to reason the same inputs from different views, and transfer the knowledge learned from one view to another through cross supervision [26]. For instance, MC-Net [31] employs different decoder networks, and conducts the consistency objective to make all decoders producing consistent predictions. Based on this, MC-Net+ [29] further expands the type of decoder networks to explore multi-view consistency. CAML [6] based on mutual learning paradigm transfers the labeled prior knowledge to unlabeled data. Nevertheless, we observe that most previous co-training methods typically lack an explicit constraint to encourage different subnets to learn complementary semantics. And we point out that such complementary information is essential to realize the potential of the multi-subnet architecture. In other words, most SS-MIS methods focus on two subnets making consistent predictions, ignoring that the complementarity of model predictions is equally crucial for learning unlabeled data.

In order to exhibit the effectiveness of cross-view complementary information, we conduct the training process analysis for different frameworks, *i.e.*, MT and co-training methods, as shown in Fig. 1.

There are three observations: (1) As shown in Fig. 2(a), limited by EMA, the performance of Teacher network is largely determined by the Student network, wasting its learning capacity. (2) As shown in Fig. 2(b), the co-training method can allow the two subnet to reason the input individually. However, due to the lack of explicit constraints, the two models gradually become consistent under cross supervision. (3) The proposed CML method encourages to make complementary model predictions. As shown in Fig. 2(c), the performance of the two subnets presents an adversarial rise, and the aggregation results benefited from cross-view complementary semantics, thereby achieving superior effectiveness.

In summary, our goal is to enable both subnets reasoning the input from different views, and the aggregated predictions can fully exploit cross-view semantics. To this end, we propose a simple yet effective cross-view mutual learning method, which is consisted of two main parts: (1) **Conflict-based feature learning (CFL)**: First, we impose a strong feature-level constraint to maximize the discrepancy between the feature extracted by two subnets, allowing two decoders to produce complementary predictions. Based on this, the cross supervision is conducted to learn more precise predictions for unlabeled data. (2) **Cross-view mutual learning (CML)**: Considering that direct cross supervision may compel two subnets to discard complementary information as Fig. 2(b), we propose to apply the CutMix operation to modify the inputs and supervisory signals. Meanwhile, we conduct a heterogeneous consistency objective, which combines two subnets' pseudo-labels, to supervise unlabeled data. The final results are obtained through aggregating

predictions from the two subnets. The proposed CML method also requires implementing the consistency objective, but we encourage the exploration of complementary semantics across different views to learn unlabeled data more effectively. The contributions of our work can be summarized as below:

- We propose a novel conflict-based feature learning (CFL) using a co-training framework, which maximizes the discrepancy between the features extracted from the two subnets, thereby enabling them to be decoded as complementary model predictions for the same input.

- We further propose a cross-view mutual learning (CML) method based on the CFL, which only requires modifying the inputs and supervisory signals for exploiting cross-view complementarity and consistency.

- The proposed CML method does not change the original network structure, and thus can be simply integrated into different segmentation models. Extensive experiments are conducted on three public datasets, demonstrating the superior segmentation performance.

## 2 Related Work

***Semi-Supervised Medical Image Segmentation.*** Many advances have been made in semi-supervised medical image segmentation over the past decade. In this field, pseudo-label supervision [2, 9, 23] and consistency regularization [13, 30, 34] are two widely-used strategies. Meanwhile, many works effectively utilize unlabeled data in diverse ways. Specifically, Yu et al. [38] utilize uncertainty information to guide student network learning highly confident targets from teacher network. Wang et al. [23] use neighbor matching to generate reliable pseudo-labels, fully exploring the embedding similarity with neighboring labeled data. Furthermore, some works promote consistency in the feature space to effectively leverage unlabeled data. In specific, Wu et al. [30] explore the pixel-level smoothness, and then encourage the inter-class separation at the feature space. Basak et al. [2] introduce a patch-based contrastive learning framework, which can impose intra-class compactness and inter-class separability. Zhang et al. [42] propose a self-aware and cross-sample prototypical learning, which employs feature-prototype similarity to enrich the diversity of prediction.

***Pseudo-Label Learning.*** In the realm of semi-supervised medical image segmentation, pseudo-labeling methods have been widely explored. These approaches [22, 27, 36] incorporate pseudo-labels of unlabeled data into the training process, augmenting the available supervisory information. Among them, a key point is how to reduce the impact of label noise [14]. Specifically, Yao et al. [36] propose a confidence-aware cross supervision network, which calculates the KL divergence between original and transformed image predictions, utilizing it as variance in their confidence-aware cross loss. Wang et al. [24] introduce a trust module to reassess pseudo-labels from model outputs, employing a threshold to select high-confidence values. Moreover, apart from integrating confidence-aware modules, some approaches focus on enhancing the quality of pseudo-labels. Li et al. [9] propose a self-ensembling strategy to construct reliable predictions through exponential moving average, mitigating noise and unstable pseudo-labels.

***Mutual Learning.*** The co-training framework is a promising method for semi-supervised medical image segmentation, typically involving two distinct subnets with similar structures but not sharing parameters [6, 25]. Particularly, they often conduct the mutual learning paradigm to transfer useful semantics from one subnet to another via pseudo-labeling, which allows them to provide different and complementary information for each other [26, 29]. Specifically, Chen et al. [5] propose a novel cross pseudo supervision, which trains two subnets with different initialization, and pseudo labels output from one perturbed segmentation network is used to supervise the other segmentation network. Zhang et al. [39] propose a semi-supervised contrastive mutual learning segmentation framework, which effectively leverages the cross-modal information and prediction consistency between different modalities to conduct contrastive mutual learning. Moreover, Wang et al. [25] further introduce a contrastive difference review module to mitigate the impact of network bias correction.

## 3 Method

In semi-supervised scenario, the training set $\mathcal{D}$ consists of a small labeled set $\mathcal{D}_l = \{(X_k^l, Y_k^l)\}_{k=1}^{N}$, and a large unlabeled set $\mathcal{D}_u = \{X_k^u\}_{k=N+1}^{M+N}$, where $N \ll M$. Specifically, $X_k \in \mathbb{R}^{H \times W \times L}$ represents the 3D medical volume and $Y_k \in \{0, 1, \ldots, C-1\}^{H \times W \times L}$ is ground truth with $C$ categories. Our goal is to train a medical image segmentation model by fully leveraging both a labeled set $\mathcal{D}_l$ and a much larger unlabeled set $\mathcal{D}_u$.

The overall pipeline of our proposed method is depicted in Fig. 3, with a co-training framework like MC-Net [31] to conduct two parallel networks to predict the supervisory signals of its counterpart. First, we conduct a novel conflict-based feature learning (CFL) paradigm that encourages the two subnets to reason the input from two different views, which will be introduced in Sec. 3.1. Second, we propose a simple yet effective cross-view mutual learning (CML) method that does not need the elaborate pseudo-label correction strategies, and instead constructs heterogeneous consistency objective to explore cross-view consistency and complementarity, as further discussed in Sec. 3.2. In the end, we detail the overall learning objective in Sec. 3.3.

### 3.1 Conflict-Based Feature Learning

In this section, we introduce a novel conflict-based feature learning (CFL) paradigm with a co-training framework. As shown in Fig. 3, we employ two subnets that share a similar architecture but have non-shared parameters to conduct mutual learning. Here, each subnet is divided into an encoder $E_i$ and a decoder $D_i$, where $i \in \{0, 1\}$, denoting the first or second subnet, respectively. Considering that the objective of CFL is to enable the two subnets to capture different features from the same input, so the outputs from different encoder networks should be distinct. In light of this, we denote the outputs from encoder networks as $f_i^\alpha = E_i(X^\alpha)$, where $\alpha \in \{l, u\}$ represents for the labeled data and the unlabeled data, respectively. Then, we impose a strong constraint $\mathcal{L}_{dis}^\alpha$ in the feature space, which minimizes the cosine similarity between the latent features $f_i^\alpha$ in a conflict learning manner. In this way, the two subnets are able to encode the same input from different views, thereby capturing different semantic information. The feature-level

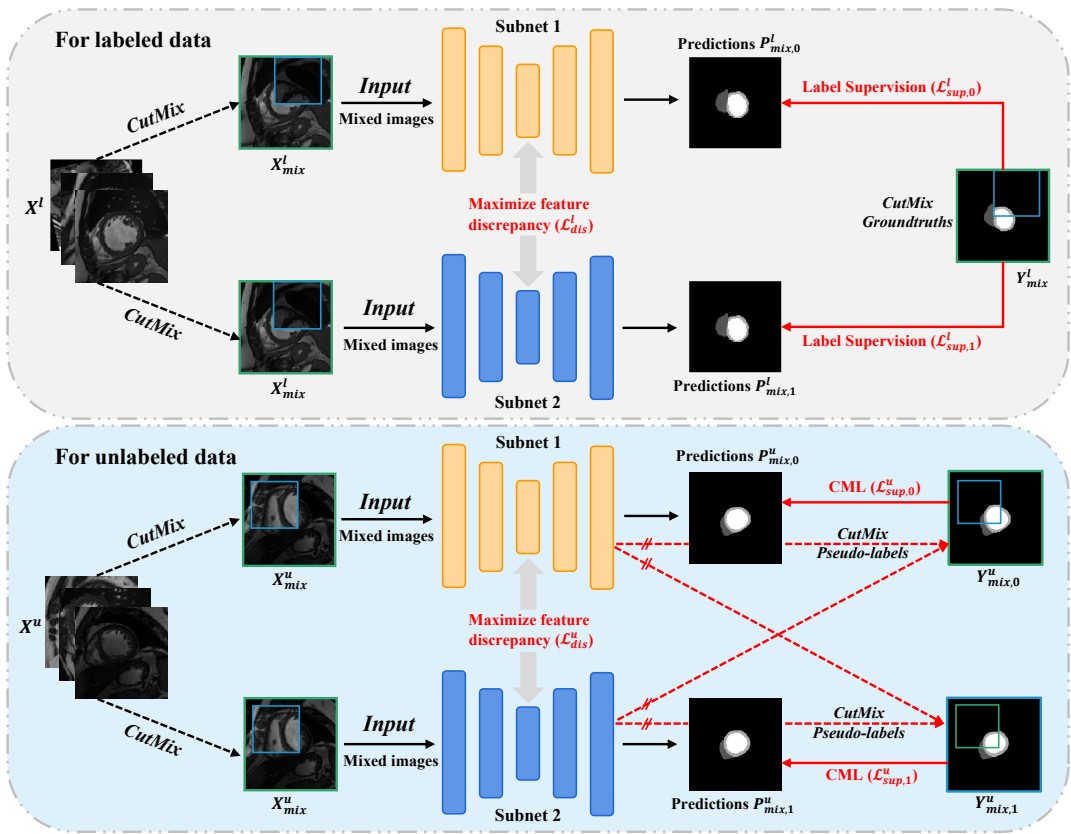

**Figure 3: Overall pipeline of CML in the co-training framework. We first apply the CutMix operation to modify the inputs and supervisory signals to conduct the supervision objective $\mathcal{L}_{sup}$. Specifically, for labeled data, we conduct the same CutMix to generate mixed labels $Y_{mix}^l$, which are used to supervise the predictions of $X_{mix}^l$. For unlabeled data, we construct heterogeneous supervisory signals $Y_{mix,0}^u$ and $Y_{mix,1}^u$, obtained by the combination of two subnets' pseudo-labels, to supervise the predictions of $X_{mix}^u$. Note that $\mathcal{L}_{dis}$ is a conflict-based unsupervised loss, aiming to learn distinct features from the same input.**

discrepancy loss $\mathcal{L}_{dis}^\alpha$ can be formulated as:

$$\mathcal{L}_{dis,i}^\alpha = 1 + \frac{f_i^\alpha \cdot \bar{f}_{(1-i)}^\alpha}{\|f_i^\alpha\| \times \|\bar{f}_{(1-i)}^\alpha\|}, \tag{1}$$

where $\bar{f}_{(1-i)}^\alpha$ is a copy of the feature $f_{(1-i)}^\alpha$ without gradients, avoiding interference with parameter updates in the other subnet. The coefficient 1 is to ensure that the value of $\mathcal{L}_{dis}^\alpha$ is always non-negative. Note that we need to conduct conflict-based feature learning on both of the two subnets, so the overall feature-level discrepancy loss can be expressed as $\mathcal{L}_{dis}^\alpha = \mathcal{L}_{dis,0}^\alpha + \mathcal{L}_{dis,1}^\alpha$.

Meanwhile, we conduct the supervision objective to enable both subnets making precise predictions. For the labeled images, we use the ground truths to supervise the model training. Specifically, the supervision objective $\mathcal{L}_{sup}^l$ is a linear combination of Dice loss and Cross-entropy loss, which can be formulated as:

$$\mathcal{L}_{sup,i}^l(P_i^l, Y^l) = \frac{1}{2}\left(\mathcal{L}_{ce}(P_i^l, Y^l) + \mathcal{L}_{dice}(P_i^l, Y^l)\right), \tag{2}$$

where $P_i^l = D_i \circ E_i(X^l)$ represents the probability outputs from the $i$-th subnet. Particularly, the ground truths are used to supervise

two subnets, and thus the supervision objective for labeled images can be defined as $\mathcal{L}_{sup}^l = \mathcal{L}_{sup,0}^l + \mathcal{L}_{sup,1}^l$.

For unlabeled images, we implement a mutual learning paradigm to guide each subnet in making reliable predictions, using pseudo-labels generated by the other subnet. Specifically, we first compute the probability outputs for unlabeled images: $P_i^u = D_i \circ E_i(X^u)$, where $i \in \{0, 1\}$ represents two different subnets. Then, the pseudo-labels are generated by *argmax* operation, i.e., $\tilde{Y}_i^u = argmax(P_i^u)$. Following pairwise pseudo-labels, we supervise one subnet's probability output with the other subnet's predictions. Then, the consistency objective for unlabeled images is:

$$\mathcal{L}_{sup}^u = \mathcal{L}_{sup,0}^u(P_0^u, \tilde{Y}_1^u) + \mathcal{L}_{sup,1}^u(P_1^u, \tilde{Y}_0^u). \tag{3}$$

In conclusion, the feature-level discrepancy loss $\mathcal{L}_{dis}$ allows the two subnets to encode different semantics from the same inputs. The supervision loss $\mathcal{L}_{sup}$ transfers useful knowledge from one subnet to another through pseudo-labeling, and thus makes more precise predictions for unlabeled data. Overall, the mutual learning objective based on CFL method is:

$$\mathcal{L} = \mathcal{L}_{sup}^l + \lambda_1 \mathcal{L}_{sup}^u + \lambda_2 \mathcal{L}_{dis}, \tag{4}$$

where $\mathcal{L}_{dis} = \mathcal{L}_{dis}^{u} + \mathcal{L}_{dis}^{l}$ is the feature-level discrepancy objective for both labeled and unlabeled images. $\lambda_1$ and $\lambda_2$ are the trade-off coefficients.

## 3.2 Cross-View Mutual Learning

Recall that the proposed CFL method in Sec. 3.1 aims to reason the same inputs from different views, but the previous mutual learning methods, *e.g.*, Eq. (3), conduct cross supervision to align the predictions of the two subnets. There are two issues: (1) Aligning the predictions of the two subnets might weaken the effectiveness of the CFL method. In other words, the two subnets might degrade to the extent where they reason the input from the same view. (2) The cross supervision paradigm would make two subnets ignoring meaningful semantics from their own view. To this end, we propose a novel cross-view mutual learning (CML) method to maximize the effectiveness of CFL. Note that the proposed CML method does not require changing the original network structure and the learning objective in Eq. (4). Instead, it only requires a simple yet effective modification of the input images and supervisory signals to achieve superior semi-supervised segmentation effectiveness.

Specifically, we first apply the CutMix operation to modify the input images. There are two advantages: (1) CutMix is a strong data augmentation that helps the two subnets further learning extra meaningful semantics, and (2) CutMix allows us to generate heterogeneous supervisory signals, which splice pseudo-labels from two subnets, to maintain own perspectives of each subnet while fully exploiting complementary semantics from the other. For labeled images, we randomly select two labeled images $(X_u^l, X_v^l)$ along with their corresponding ground truths $(Y_u^l, Y_v^l)$. Then, we paste a random crop from $\mathbf{X}_v^l$ into labeled image $\mathbf{X}_u^l$. The mixed image-label pairs for labeled data can be expressed as:

$$X_{mix}^l = X_u^l \odot \mathcal{M} + X_v^l \odot (1 - \mathcal{M}), \tag{5}$$

$$Y_{mix}^l = Y_u^l \odot \mathcal{M} + Y_v^l \odot (1 - \mathcal{M}), \tag{6}$$

where $\mathbf{1} \in \{1\}^{H \times W \times L}$, and $\mathcal{M} \in \{0,1\}^{H \times W \times L}$ denotes a zero-centered mask, where the size of the zero-value region is $\beta H \times \beta W \times \beta L$, with $\beta \in (0,1)$. The symbol $\odot$ represents element-wise multiplication.

Intuitively, we have no need to adjust the ground truths to supervise labeled images, so the mixed label $Y_{mix}^l$ is conducted to directly supervise the model training. The supervision objective for labeled images can be redefined as:

$$\mathcal{L}_{sup}^l = \mathcal{L}_{sup,0}^l(P_{mix,0}^l, Y_{mix}^l) + \mathcal{L}_{sup,1}^l(P_{mix,1}^l, Y_{mix}^l), \tag{7}$$

where $P_{mix,i}^l = D_i \circ E_i(X_{mix}^l)$ represents the probability outputs from the $i$-th network. $i \in \{0,1\}$ denotes two different subnets.

Considering that the CFL method in Eq. (1) makes the two subnets reasoning the same inputs from different views, the outputs are thus complementary. In light of this, we aim to modify the supervisory signals for unlabeled images, which contain both pseudo-labels from the own perspective and complementary predictions from the other subnet. In specific, we apply the similar CutMix operation in Eq. (5) for unlabeled inputs. Differently, we implement heterogeneous mutual learning to make the two subnets acquiring useful semantics from unlabeled images. Given two unlabeled images

$(X_p^u, X_q^u)$, the mixed image-label pairs for unlabeled data can be expressed as:

$$X_{mix}^u = X_p^u \odot \mathcal{M} + X_q^u \odot (1 - \mathcal{M}), \tag{8}$$

$$Y_{mix,i}^u = \hat{Y}_{p,(1-i)}^u \odot \mathcal{M} + \hat{Y}_{q,i}^u \odot (1 - \mathcal{M}), \tag{9}$$

where $\hat{Y}_{p,(1-i)}^u$ and $\hat{Y}_{q,i}^u$ are pseudo-labels for the unlabeled images $X_p^u$ and $X_q^u$, obtained from the $(1-i)$-th subnet and the $i$-th subnet, respectively. Taking $\hat{Y}_{q,i}^u$ as example, we first input $X_q^u$ to the $i$-th subnet for predicting its probability output: $P_{q,i}^u = D_i \circ E_i(X_q^u)$. Then, we conduct *argmax* operation: $\tilde{Y}_{q,i}^u = argmax(P_{q,i}^u)$, and employ non-maximum suppression (NMS) [4] on $\tilde{Y}_{q,i}^u$ to get the final pseudo-labels $\hat{Y}_{q,i}^u$. Note that we conduct the same operation for unlabeled image $X_p^u$ to generate the pseudo-labels $\hat{Y}_{p,(1-i)}^u$.

Following pairwise heterogeneous pseudo-labels in Eq. (9), we implement the consistency objective to guide each subnet in capturing useful semantics from unlabeled images. Since we employ a mixed and different supervisory signal, the two subnets are capable of fully exploiting the cross-view complementary information from the CFL method in Sec. 3.1, while maintaining that of their own perspectives. Overall, the consistency objective for unlabeled images can be redefined as:

$$\mathcal{L}_{sup}^u = \mathcal{L}_{sup,0}^u(P_{mix,0}^u, Y_{mix,0}^u) + \mathcal{L}_{sup,1}^u(P_{mix,1}^u, Y_{mix,1}^u), \tag{10}$$

where $P_{mix,i}^u = D_i \circ E_i(X_{mix}^u)$ represents the probability outputs from the $i$-th network.

Since the feature-level discrepancy loss $\mathcal{L}_{dis}$ is an unsupervised loss to maximize discrepancy in feature space, it is not related to the supervisory signals. Hence, we do not need to change the form of $\mathcal{L}_{dis}$. Note that $f_i^\alpha = E_i(X_{mix}^\alpha)$ in the CML method, where $\alpha \in \{l, u\}$ represents for the labeled data and the unlabeled data, respectively. Besides, the proposed CML method only changes the input images and supervisory signals, and thus the overall learning objective is in form equal to Eq. (4), using the redefined supervision objective $\mathcal{L}_{sup}^l$ and $\mathcal{L}_{sup}^u$ in Eqs. (7, 10).

## 3.3 The Overall Learning Objective

Inspired by previous work [1], we apply the CutMix augmentation on labeled data for training a supervised model during pre-training. Meanwhile, we conduct the feature-level discrepancy objective to encourage the two subnets capturing different semantics from the same inputs. Then, the entire learning objective during pre-training can be formulated as:

$$\mathcal{L}_{pre} = \mathcal{L}_{sup}^l + \lambda_2 \mathcal{L}_{dis}^l, \tag{11}$$

where $\lambda_2$ is a trade-off coefficient in Eq. (4).

During self-training, we initialize two subnets using the pre-trained weights from Eq. (11). Then, we jointly learn the supervision objectives $\mathcal{L}_{sup}^l$ and $\mathcal{L}_{sup}^u$, and feature-level discrepancy objective $\mathcal{L}_{dis}$, the total loss is in form equal to Eq. (4). Note that we redefine $\mathcal{L}_{sup}^l$ and $\mathcal{L}_{sup}^u$ in Sec. 3.2 to fully exploit cross-view semantics.

In the testing stage, given a test image $X_{test}$, we first compute the probability maps from the two subnets: $P_{test,i} = D_i \circ E_i(X_{test})$. And the final prediction maps can be obtained in an equal-sum manner by: $Y_{test} = argmax((P_{test,0} + P_{test,1})/2)$.

**Table 1: Comparison results with SoTA semi-supervised segmentation methods on the LA dataset.**

| Method | Scans used | | Metrics | | |
| --- | --- | --- | --- | --- | --- |
| | Labeled | Unlabeled | Dice(%)↑ | 95HD(voxel)↓ | ASD(voxel)↓ |
| V-Net | 4(5%) | 0 | 52.55 | 47.05 | 9.87 |
| V-Net | 8(10%) | 0 | 82.74 | 13.35 | 3.26 |
| V-Net | 80(All) | 0 | 91.47 | 5.48 | 1.51 |
| UA-MT[38] | | | 82.26 | 13.71 | 3.82 |
| SASSNet[10] | | | 81.60 | 16.16 | 3.58 |
| DTC[13] | | | 81.25 | 14.90 | 3.99 |
| URPC[15] | 4(5%) | 76(95%) | 82.48 | 14.65 | 3.65 |
| MC-Net[31] | | | 83.59 | 14.07 | 2.70 |
| SS-Net[30] | | | 86.33 | 9.97 | 2.31 |
| MCF[25] | | | 86.52 | 9.12 | 2.40 |
| Ours | | | **87.63** | **8.92** | **2.23** |
| UA-MT[38] | | | 87.79 | 8.68 | 2.12 |
| SASSNet[10] | | | 87.54 | 9.84 | 2.59 |
| DTC[13] | | | 87.51 | 8.23 | 2.36 |
| URPC[15] | 8(10%) | 72(90%) | 86.92 | 11.13 | 2.28 |
| MC-Net[31] | | | 87.62 | 10.03 | 1.82 |
| SS-Net[30] | | | 88.55 | 7.49 | 1.90 |
| MCF[25] | | | 88.05 | 8.32 | 2.08 |
| Ours | | | **90.36** | **6.06** | **1.68** |

**Table 2: Comparison results with SoTA semi-supervised segmentation methods on the ACDC dataset.**

| Method | Scans used | | Metrics | | |
| --- | --- | --- | --- | --- | --- |
| | Labeled | Unlabeled | Dice(%)↑ | 95HD(voxel)↓ | ASD(voxel)↓ |
| U-Net | 3(5%) | 0 | 47.83 | 31.16 | 12.62 |
| U-Net | 7(10%) | 0 | 79.41 | 9.35 | 2.70 |
| U-Net | 70(All) | 0 | 91.44 | 4.30 | 0.99 |
| UA-MT[38] | | | 46.04 | 20.08 | 7.75 |
| SASSNet[10] | | | 57.77 | 20.05 | 6.06 |
| DTC[13] | | | 56.90 | 23.36 | 7.39 |
| URPC[15] | 3(5%) | 67(95%) | 55.87 | 13.60 | 3.74 |
| MC-Net[31] | | | 62.85 | 7.62 | 2.33 |
| SS-Net[30] | | | 65.82 | 6.67 | 2.28 |
| MCF[25] | | | 82.37 | 5.74 | 1.59 |
| Ours | | | **88.53** | **1.87** | **0.57** |
| UA-MT[38] | | | 81.65 | 6.88 | 2.02 |
| SASSNet[10] | | | 84.50 | 5.42 | 1.86 |
| DTC[13] | | | 84.29 | 12.81 | 4.01 |
| URPC[15] | 7(10%) | 63(90%) | 83.10 | 4.84 | 1.53 |
| MC-Net[31] | | | 86.44 | 5.50 | 1.84 |
| SS-Net[30] | | | 86.78 | 6.07 | 1.40 |
| MCF[25] | | | 87.67 | 3.89 | 1.14 |
| Ours | | | **89.42** | **1.42** | **0.52** |

## 4 Experiments

### 4.1 Datasets and Evaluation Metrics

**LA dataset.** The LA dataset [33] is the benchmark dataset for the 2018 Atrial Segmentation Challenge, which comprises 100 gadolinium-enhanced MR imaging scans (GE-MRIs) and corresponding ground truths, with an isotropic resolution of $0.625 \times 0.625 \times 0.625$ mm$^3$.

**ACDC dataset.** The ACDC dataset [3] is the benchmark dataset for the Automated Cardiac Diagnosis Challenge, which contains 100 short axis cine-MRIs, and expert annotations are provided for three classes: left and right ventricle (LV, RV), and myocardium (MYO).

**BraTS2019 dataset.** The BraTS2019 dataset [16] is a whole brain tumor segmentation dataset, which contains 335 scans with four modalities (FLAIR, T1, T1ce, and T2), and each sequence with an isotropic resolution of 1 mm$^3$.

**Metrics.** Three widely-used metrics are used to evaluate the model performance, including Dice score (%), 95% Hausdorff Distance (95HD) in voxel, and Average Surface Distance (ASD) in voxel.

### 4.2 Implementation Details

The proposed CML method is implemented in PyTorch and executed on an NVIDIA GeForce RTX 3090 GPU. Following previous works [13, 15, 30], we apply random cropping, flipping, and rotation to augment the training dataset. All the segmentation tasks are optimized using an SGD optimizer with an initial learning rate of 0.01. **For LA and BraTS2019 datasets**, we set $\lambda_1 = 1$ and $\lambda_2 = 0.2$ to implement cross-view mutual learning. Inspired by [1], the zero-centered mask $\mathcal{M}$ is used for CutMix operation, where we set $\beta = \frac{2}{3}$. Additionally, we follow [15, 30] to randomly crop $112 \times 112 \times 80$ patches for the LA dataset, and $96 \times 96 \times 96$ for the BraTS2019 dataset. The 3D V-Net is chosen as the backbone, and the batch size is 8, including four labeled patches and four unlabeled patches. Particularly, the pre-training and self-training iterations on LA dataset

**Table 3: Comparison results with SoTA semi-supervised segmentation methods on the BraTS2019 dataset.**

| Method | Scans used | | Metrics | | |
| --- | --- | --- | --- | --- | --- |
| | Labeled | Unlabeled | Dice(%)↑ | 95HD(voxel)↓ | ASD(voxel)↓ |
| V-Net | 25(10%) | 0 | 78.32 | 22.29 | 7.36 |
| V-Net | 50(20%) | 0 | 80.18 | 20.57 | 6.09 |
| V-Net | 250(All) | 0 | 88.23 | 7.21 | 1.53 |
| MT[19] | | | 81.70 | 13.28 | 3.56 |
| DAN[41] | | | 82.50 | 15.11 | 3.79 |
| UA-MT[38] | | | 80.93 | 17.71 | 5.43 |
| ICT[20] | 25(10%) | 225(90%) | 82.70 | 13.43 | 4.07 |
| EM[21] | | | 82.35 | 14.70 | 3.68 |
| URPC[15] | | | 84.16 | 11.01 | 2.63 |
| MCF[25] | | | 83.67 | 12.58 | 3.28 |
| Ours | | | **85.26** | **9.08** | **1.83** |
| MT[19] | | | 85.03 | **7.80** | 1.89 |
| DAN[41] | | | 84.63 | 8.96 | 2.34 |
| UA-MT[38] | | | 85.05 | 12.31 | 3.03 |
| ICT[20] | 50(20%) | 200(80%) | 84.67 | 8.97 | 2.39 |
| EM[21] | | | 84.82 | 12.37 | 3.21 |
| URPC[15] | | | 85.49 | 8.47 | 2.04 |
| MCF[25] | | | 84.85 | 11.24 | 2.29 |
| Ours | | | **86.63** | 7.83 | **1.45** |

are set as 2k and 15k, and on BraTS2019 dataset are set as 10k and 30k. **For ACDC dataset**, we set $\lambda_1 = 0.5$, $\lambda_2 = 0.1$ for CML, and $\beta = \frac{1}{2}$ for CutMix operation. Following [30], we use a 2D U-Net as the backbone. The input patch size is set as $256 \times 256$ (2D slices), and the batch size is 24. The iterations of pre-training and self-training are set as 10k and 30k, respectively.

### 4.3 Comparisons and Results with SoTA

In this section, we conduct experiments with varying ratios of labeled data, and compare the proposed CML with other SSMIS methods. Specifically, **for LA and ACDC datasets**, we evaluate

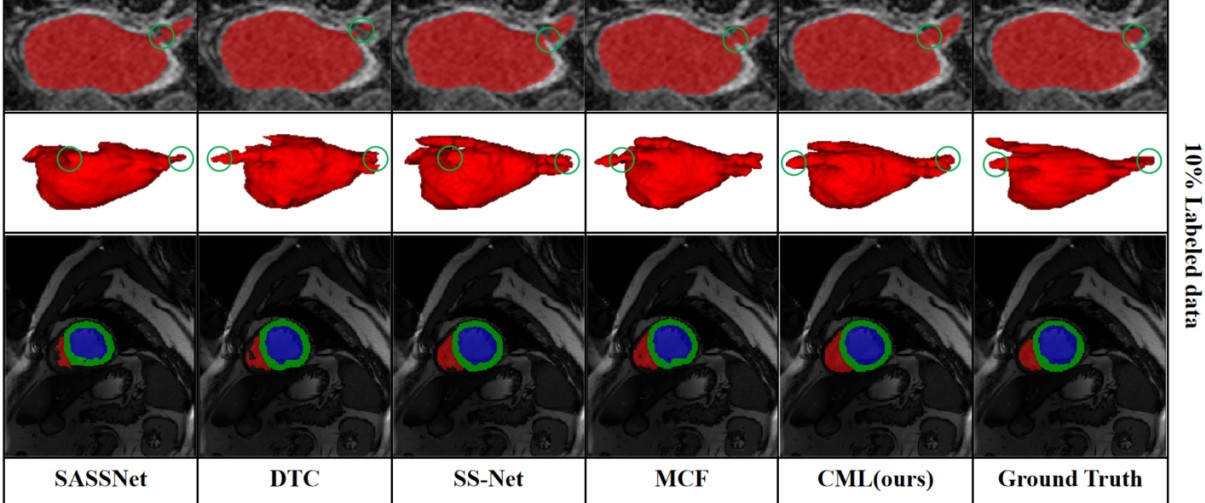

**Figure 4: Segmentation visualization of different SSMIS methods with 10% labeled data on the LA and ACDC datasets.**

**Table 4: Ablation studies among different losses on the LA and ACDC datasets with 10% labeled data.**

| $\mathcal{L}_{sup}^{l}$ | $\mathcal{L}_{dis}$ | $\mathcal{L}_{sup}^{u}$ | LA dataset | | |
|:---:|:---:|:---:|:---:|:---:|:---:|
| | | | Dice(%)↑ | 95HD(voxel)↓ | ASD(voxel)↓ |
| ✓ | | | 82.74 | 13.35 | 3.26 |
| ✓ | ✓ | | 83.73 | 18.92 | 4.80 |
| ✓ | | ✓ | 88.16 | 9.83 | 2.32 |
| ✓ | ✓ | ✓ | **90.36** | **6.06** | **1.68** |

| $\mathcal{L}_{sup}^{l}$ | $\mathcal{L}_{dis}$ | $\mathcal{L}_{sup}^{u}$ | ACDC dataset | | |
|:---:|:---:|:---:|:---:|:---:|:---:|
| | | | Dice(%)↑ | 95HD(voxel)↓ | ASD(voxel)↓ |
| ✓ | | | 79.41 | 9.35 | 2.70 |
| ✓ | ✓ | | 80.27 | 7.69 | 2.36 |
| ✓ | | ✓ | 87.75 | 6.35 | 1.71 |
| ✓ | ✓ | ✓ | **89.42** | **1.42** | **0.52** |

our method against seven SoTA methods, including UA-MT [38], SASSNet [10], DTC [13], URPC [15], MC-Net [31], SS-Net [30], and MCF [25]. To ensure a fair comparison, we report the performance of all competitors under identical experimental settings in SS-Net [30] across different labeled ratios (*i.e.*, 5% and 10%).

Tabs. 1 and 2 show that the proposed CML method outperforms previous SSMIS methods, and achieves results very close to the fully-supervised counterpart. There are three observations: (1) Most previous methods, like UA-MT, effectively exploit unlabeled data with a Mean Teacher framework. However, the model performance is limited by EMA, wasting the potential of multi-subnet architecture. The proposed CML method significantly improves Dice by 5.37% and 2.57% compared to UA-MT on LA dataset with 5% and 10% labeled data. (2) MC-Net and MCF exploit unlabeled data with a co-training framework, where they conduct the mutual learning to make two subnets producing consistent predictions, which are

close to our approach. However, they lack an explicit constraint to explore the complementarity of model predictions. In contrast, CML introduces a new conflict-based feature learning, which encourages the two subnets reasoning the same input from different views. Notably, the proposed CML improves 95HD by 4.08 and 2.47, compared to MC-Net and MCF on ACDC dataset with 10% labeled data. (3) The CML method can fully exploit cross-view semantics to make more precise predictions for unlabeled data. As presented in Fig. 4, our method excels in accurately segmenting intricate details of the target organ, particularly in cases where edges are prone to misidentification, as highlighted by green circles.

**For BraTS2019 dataset**, we compare the proposed CML method with MT [19], UA-MT [38], ICT [20], EM [21], URPC [15], and MCF [25] in Tab. 3. All methods strictly follow the experimental setting in URPC [15] using 10% and 20% labeled data. Similarly, Our CML outperforms the previous SSMIS methods on the BraTS2019 dataset. In particular, URPC employs uncertainty estimation to implement multi-layer consistency learning. However, they ignore that the complementarity of model predictions is equally crucial, which would limit the SSMIS performance. The proposed CML achieves significant improvement in Dice, 95HD, and ASD terms, surpassing the URPC by 1.1%, 1.93, and 0.8 under 10% labeled data.

### 4.4 Ablation for Loss Components

To understand the effectiveness of loss components in CML, we remove each loss individually to observe the corresponding changes in performance. Specifically, (A) $\mathcal{L}_{sup}^{l}$ is a supervision loss to learn labeled data, preventing model collapse. (B) $\mathcal{L}_{dis}$ is a feature-level discrepancy loss that encourages the two subnets to reason the same input from different views. (C) $\mathcal{L}_{sup}^{u}$ is a heterogeneous consistency loss to learn unlabeled data, fully exploring the complementarity of model predictions. As shown in Tab. 4, $\mathcal{L}_{sup}^{u}$ plays a crucial role in the proposed CML method, effectively learning useful semantics from unlabeled data through cross-view mutual learning. Under 10% labeled data, the model with $\mathcal{L}_{sup}^{u}$ improves 5.42% and 8.34% in

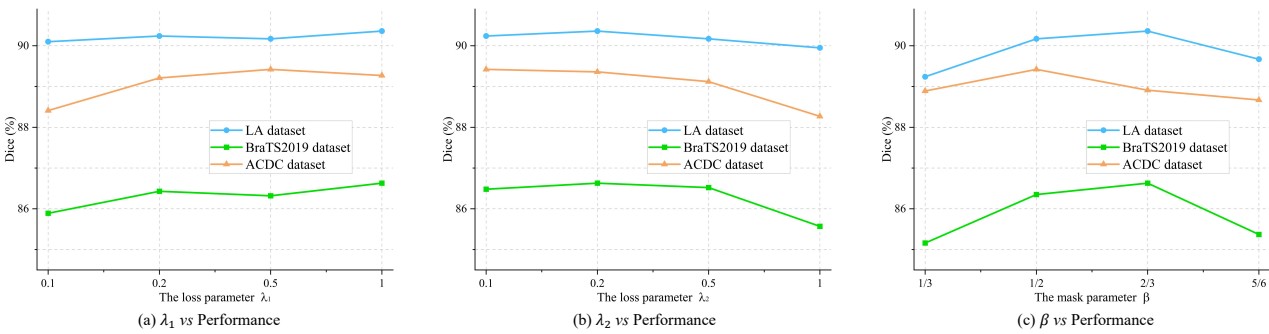

**Figure 5: The model parameter sensitivity analysis for $\lambda_1$, $\lambda_2$, and $\beta$ on three public datasets using 10% labeled data.**

Dice term compared to single $\mathcal{L}^l_{sup}$ on the ACDC and LA datasets, respectively. Moreover, $\mathcal{L}_{dis}$ ensures that different encoders output distinct features, and thus capture cross-view complementary semantics. Specifically, $\mathcal{L}_{dis}$ further improves 2.2% and 1.67% in Dice term on the ACDC and LA datasets, respectively. Notably, the complete CML further achieves superior edge segmentation performance, improving the model without $\mathcal{L}_{dis}$ by 3.77 in 95HD term, 0.64 in ASD term on LA dataset, and 4.93 in 95HD term, 1.19 in ASD term on ACDC dataset, respectively.

## 4.5 Ablation for Model Parameters

In this section, we conduct the model parameter sensitivity analysis for $\lambda_1$, $\lambda_2$, and $\beta$. In detail, $\lambda_1$ and $\lambda_2$ are the balance weights used to control the influence of different loss compositions. We set the hyperparameters $\lambda_1$ and $\lambda_2$ in the range of {0.1, 0.2, 0.5, 1}. As shown in Figs. 5(a) and 5(b), the proposed CML is not sensitive when $\lambda_1$ changes from 0.2 to 1, and $\lambda_2$ changes from 0.1 to 0.5. Empirically, we set $\lambda_1 = 1$ and $\lambda_2 = 0.2$ for LA and BraTS2019 datasets, and $\lambda_1 = 0.5$ and $\lambda_2 = 0.1$ for ACDC dataset.

In addition, we further explore the optimal setting for zero-centered mask $\mathcal{M}$, *i.e.*, the parameter $\beta$. Specifically, we set the hyperparameters $\beta = \{\frac{1}{3}, \frac{1}{2}, \frac{2}{3}, \frac{5}{6}\}$ to observe how the performance changes. As depicted in Fig. 5(c), the performance of the model exhibits a clear decline in Dice score when $\beta$ is set as a larger or smaller value. This might be because a larger or smaller zero-centerd mask could hinder the effectiveness of CutMix operation, making it difficult to exploit the heterogeneous consistency objective to transfer useful semantics from one subnet to another.

## 4.6 Effectiveness of CML

Recalling our proposed method, we first suggest employing co-training framework to release the potential of multi-subnet architectures, where the cross pseudo-supervision is used to learn from unlabeled data, denoted as CPS. Second, we introduce a new conflict-based feature learning (CFL), which imposes a strong feature-level constraint based on CPS to encourage the two subnets reasoning about the same input from different views. In the end, building upon CFL, we further propose to implement a heterogeneous consistency objective through cross-view mutual learning (CML), fully exploring the complementarity of model predictions.

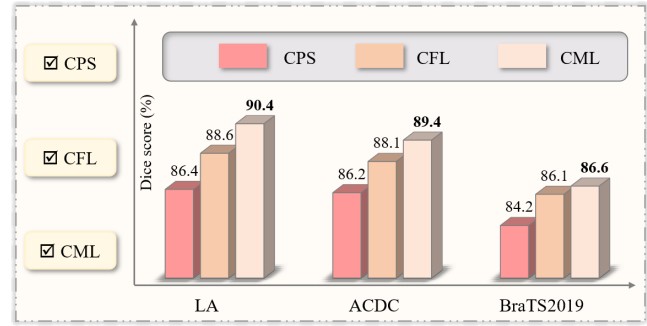

**Figure 6: The comparison experiment among CPS, CFL, and CML methods on three dataset under 10% labeled data.**

To evaluate the effectiveness of CML, we conduct a comparison experiment among CPS, CFL, and CML methods with a co-training framework. As shown in Fig. 6, CML achieves superior performance under 10% labeled data. Particularly, for the LA and ACDC datasets, CML with the CutMix operation improves Dice score by 1.8% and 1.3% compared to CFL. This is because the CutMix operation can effectively enrich the sample space for medical image datasets. Meanwhile, the proposed CML aggregates the complementary predictions from two subnets, thereby learning unlabeled data robustly. Overall, both the proposed CFL and CML methods contribute to improving the performance of co-training framework.

## 5 Conclusion

In this work, we propose a novel cross-view mutual learning (CML) method for semi-supervised medical image segmentation. Specifically, we first introduce a conflict-based feature learning (CFL) paradigm that imposes a strong feature-level constraint to encourage both subnets encoding cross-view complementary semantics. Building upon CFL, we further employ the CutMix operation to construct a heterogeneous consistency objective, allowing us to fully explore the complementarity of model predictions. Note that the proposed CML method does not alter the origin network, and thus can be simply integrated into different segmentation models. Extensive experiments conducted on three datasets demonstrate the SoTA segmentation performance of CML.

# Acknowledgments

This work was supported in part by Sichuan Science and Technology Program (No. 2024NSFSC1473), and in part by Shenzhen Science and Technology Program (Nos. JCYJ20230807115959041 and JCYJ20230807120010021).

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
