# OpenReview forum: "Cross-View Mutual Learning for Semi-Supervised Medical Image Segmentation"
_acmmm.org/ACMMM/2024/Conference — MM2024 Poster_

### Official Review · Reviewer_LV7p · 2024-05-03

**Rating:** 4
**Confidence:** 3

**Summary:**

This paper introduces a novel approach to semi-supervised medical image segmentation (SSMIS) by leveraging a multi-subnet architecture with cross-view mutual learning. Traditional methods in SSMIS focus on consistency between subnets' predictions for unlabeled data but overlook the importance of complementarity in model predictions.

The proposed method consists of two key components: conflict-based feature learning (CFL) and cross-view mutual learning (CML). CFL encourages each subnet to learn distinct features from the same input, which are then decoded into complementary model predictions. This allows both subnets to understand the input from different perspectives. CML maximizes the effectiveness of CFL by implementing a heterogeneous consistency objective, which exploits the complementarity of model predictions across different views.

**Strengths:**

1. The writing is excellent—clear, concise, and easy to understand.
2. It's reasonable to emphasis the importance of complementarity in model predictions. This approach, which enhances the Conflict-Based Feature Learning (CFL) paradigm, encourages each subnet to interpret the input from unique perspectives, fostering more robust and diverse feature extraction.

**Limitations:**

1. In figure1, the demonstration of (b) and (c) seem to be similar, it would be better to emphasis the distinction of proposed CML framework.
2. The introduction to the related work on semi-supervised segmentation could be more comprehensive. These studies represent different branches of semi-supervision for medical image segmentation:
[1] Peng J, Wang P, Desrosiers C, et al. Self-paced contrastive learning for semi-supervised medical image segmentation with meta-labels[J]. Advances in Neural Information Processing Systems, 2021, 34: 16686-16699.
[2] Zeng D, Wu Y, Hu X, et al. Positional contrastive learning for volumetric medical image segmentation[C]//Medical Image Computing and Computer Assisted Intervention–MICCAI 2021: 24th International Conference, Strasbourg, France, September 27–October 1, 2021, Proceedings, Part II 24. Springer International Publishing, 2021: 221-230.
[3] Wu Y, Chen J, Yan J, et al. GCL: Gradient-Guided Contrastive Learning for Medical Image Segmentation with Multi-Perspective Meta Labels[C]//Proceedings of the 31st ACM International Conference on Multimedia. 2023: 463-471.

**Suitability:**

2

---

### Official Review · Reviewer_ECoC · 2024-05-24

**Rating:** 5
**Confidence:** 3

**Summary:**

This work addresses the discrepancies among networks and proposes a cross-view mutual learning approach to maximize the effectiveness of conflict-based feature learning. This ensures that the aggregated predictions capture both consistency and complementarity across all views. Extensive experiments demonstrate the effectiveness of the proposed method.

**Strengths:**

1.The paper is well-written and is with good organization.
2.Unlike previous works that focus on achieving consistent results from two subnetworks, the authors propose to focus on the complementarity of model predictions, which is an interesting idea.
3.Compared to existing state-of-the-art methods, the proposed CML method achieves excellent performance.

**Limitations:**

1.	The model has three hyperparameters. Finding the optimal settings for these parameters might require additional tuning.
2.	Regarding the experimental evaluation, please clarify the working principles of the comparison methods to better highlight the contribution of the proposed method.

**Suitability:**

3

---

### Official Review · Reviewer_7KNC · 2024-05-24

**Rating:** 5
**Confidence:** 3

**Summary:**

To address the issue that existing Semi-supervised medical image segmentation (SSMIS) often ignore that the complementarity of model predictions is equally crucial, this paper imposes a strong feature-level constraint to make the two subnets producing the complementary predictions. A novel conflict-based feature learning (CFL) method and a cross-view mutual learning (CML) method based on CFL are proposed.

**Strengths:**

1) The complementarity of model predictions is an important factor for semi-supervised learning. The authors have fully explored cross-view semantic information from both feature-level and image-level dimensions, thus the model can learn unlabeled data effectively.
2) Extensive experimental results demonstrate the superiority of the proposed CML method. The code is also provided.

**Limitations:**

CutMix is a strong data augmentation method that can improve the performance of semi-supervised learning to some extent. Although the authors innovatively use CutMix to construct heterogeneous supervision objective, a comparison with a CutMix-only paradigm is needed to illustrate the effectiveness of CML.

**Suitability:**

3

---

### Official Review · Reviewer_fivy · 2024-05-24

**Rating:** 4
**Confidence:** 3

**Summary:**

This paper proposes a novel Cross-view Mutual Learning (CML) method with a two-branch co-training framework to realize the potential of the multi-subnet architecture. Specifically, CML encourages learning different latent features from two subnets, thereby producing complementary predictions. Meanwhile, CML constructs heterogeneous consistency objective via CutMix operation to explore cross-view semantics.

**Strengths:**

(1) The authors detail the inability of previous works to realize the potential of multi-subnet architectures and conduct comparative experiments. This observation is reasonable and, based on it, the proposed CFL is used to learn cross-view semantics.
(2) The use of CutMix operation is reasonable. The authors employ CutMix to construct heterogeneous supervisory signals, effectively preventing the two subnets from degrading to reasoning from the same perspective.
(3) The proposed CML achieves superior performance on three publicly available datasets, and the authors release their codes and datasets.

**Limitations:**

(1) CutMix is a popular data enhancement technique and is also used in the semi-supervised domain. What are the differences between the proposed CML and previous works?
(2) The proposed conflict-based feature learning (CFL) encourages the model to learn distinct features from different views by introducing adversarial loss. May this adversarial loss interfere with other loss terms, and lead to model degradation problem?

**Suitability:**

2

---

### Meta-Review · Area_Chair_EV8X · 2024-07-06

**Recommendation:** Accept (Poster)
**Confidence:** 2

**Metareview:**

The paper introduces a conflict-based feature learning (CFL) paradigm and a cross-view mutual learning (CML) method, enhancing the performance of semi-supervised medical image segmentation. Reviewers highlighted its strengths in exploring cross-view semantics, effectively using CutMix, and achieving improved performance on public datasets. Concerns about the comparison with CutMix-only paradigms and potential model degradation due to adversarial loss were discussed in the rebuttal. Although the reviewers' discussions on the paper's limitations were somewhat brief, they unanimously leaned towards acceptance, acknowledging the contributions and clear writing. Considering the paper's incremental contributions and improved performance, I decide to follow the reviewers' recommendations.